# Risk assessment of arteriovenous fistulas focusing on the relationships between the properties of shunted blood flow sounds and a medical/surgical history of hemodialysis patients

Keiko Tanaka[1][¤][*], Keisuke Nishijima[2], Ken'ichi Furuya[2], Satoko Shin[3], Michiaki Kai[4]

**1** Graduate School Doctoral Program, Graduate School of Nursing, Department of Nursing, Oita University of Nursing and Health Sciences, Oita, Japan, **2** Division of Computer Science and Intelligent Systems, Faculty of Science and Technology, Oita University, Oita, Japan, **3** Fundamental Nursing, Department of Basic Nursing Sciences, Oita University of Nursing and Health Sciences, Oita, Japan, **4** Department of Health Science, School of Health Science, Nippon Bunri University, Oita, Japan

¤Current Address: Graduate School Doctoral Program, Graduate School of Nursing, Department of Nursing, Oita University of Nursing and Health Sciences, Oita, Japan

* ktanaka@oita-nhs.ac.jp

## Abstract

The global prevalence of end-stage kidney failure is increasing, with hemodialysis as the primary treatment. An arteriovenous fistula serves as a critical lifeline for patients undergoing hemodialysis, yet its function often deteriorates due to complications such as narrowing or blockage of the blood vessels. This study aimed to clarify the relationships between the sound properties of blood flow through arteriovenous fistulas and a medical/surgical history of patients undergoing hemodialysis by analyzing the distribution of sound frequencies from 100 to 4,000 Hz. Data were collected from 53 patients to identify two key parameters: the time point within one cycle of arteriovenous fistula sounds where the power distribution reached its peak, expressed as a percentage, and the specific frequency where the power was highest within the analyzed range. The results showed that well-functioning arteriovenous fistulas exhibited peak power within the first 25 percent of the sound cycle and the highest power at 200 Hz. In contrast, higher peak percentages and lower power at 200 Hz were associated with surgical interventions due to complications such as narrowing or blockage of the arteriovenous fistula. These findings suggest that the sound properties of arteriovenous fistulas, combined with patient-specific characteristics, may serve as non-invasive indicators of arteriovenous fistula function and help predict the risk of complications. This approach provides valuable insights for improving the management of arteriovenous fistulas and patient outcomes in hemodialysis therapy.

**Data availability statement:** The data are available as supplementary files for this paper (S1 Data to S6 Data files).

**Funding:** This work was supported by JSPS KAKENHI Grant Numbers JP17K17415, JP21K10775. The funders had no role in study design, data collection and analysis, decision to publish, or preparation of the manuscript.

**Competing interests:** The authors have declared that no competing interests exist.

## Introduction

Globally, more than 850 million people have kidney diseases [1], and the prevalence of end-stage renal failure continues to rise. In Asian countries such as Taiwan and Brunei, as well as in the United States, more than 400 persons per million population are diagnosed with kidney disease annually. Consequently, the number of patients undergoing hemodialysis (HD), the primary treatment for end-stage renal failure, continues to increase each year [2]. The prevalence of dialysis in Japan is 2,786 per million people, the second highest in the world. Of these patients, 90% undergo HD, which necessitates the creation of an arteriovenous fistula (AVF) [3]. With an AVF, HD serves as a kidney replacement therapy, acting as a critical lifeline for patients with end-stage renal failure. Effective self-management, supported by physicians and nurses, is crucial to maintaining the optimal HD condition over time. However, AVFs often fail due to the direct inflow of blood from arteries to veins, placing significant strain on the vessels. Consequently, the primary patency rate of AVFs is about 50–60% within 3–5 years after AVF creation [4,5]. When AVF failure occurs, percutaneous transluminal angioplasty, thrombus aspiration, or vascular stenting procedures are performed. Ongoing research aims to develop safer and more efficient methods for these interventions [6,7]. However, if the vessel cannot be adequately dilated, AVF reconstruction becomes necessary [8]. To address the needs of patients requiring AVF reconstruction, research is ongoing to develop safer and more effective techniques with higher success rates [9]. Each treatment session imposes both physical and financial burdens on a person with end-stage renal failure. Therefore, it is essential to enhance self-management skills and facilitate the early detection of abnormalities before the need for endovascular treatment or AVF reconstruction arises. This approach enables long term HD using the initially created AVF and alleviates the physical and financial burdens for people with end-stage renal failure. Various methods, including ultrasonography, blood tests, and measurement of dialysis volume during HD [10,11], have been explored for the early detection of AVF abnormalities. However, these methods are only available when patients visit a hospital for HD, making it challenging to perform them on a daily basis.

Auscultation of AVF blood flow sounds (AVF sounds) is one method for patients to self-monitor the condition of their AVF at home. Auscultation is a simple and cost-effective method that requires only a stethoscope, and many HD patients receive instruction on how to perform auscultation of AVF sounds. However, most people with end-stage renal failure are aged 65 years or older [2], and one in three individuals aged 65–74 years, as well as approximately half of those aged 75 years or older, have hearing loss [12]. Thus, auscultation as a form of AVF self-care becomes increasingly challenging with age. In addition, auscultation is subjective and challenging to evaluate objectively. AVF sounds, generated by the blood flow from arteries to veins within an AVF can be objectively analyzed through frequency analysis. The intensity of AVF sounds is anticipated to increase during the systolic phase of the cardiac cycle and decrease during diastole, reflecting variations in blood flow and blood flow velocity. Therefore, any alterations in vessel wall resistance or abnormal blood flow may be reflected in the blood flow sounds. In fact, previous studies have

demonstrated that stenosis in the AVF generates characteristic features in the high-frequency band [13–15]. In other words, we hypothesize that analyzing the frequency properties of AVF sounds facilitates the functional assessment of AVFs, thereby informing treatment strategies and nursing interventions to support long-term HD. End-stage renal failure is caused by several diseases, including hypertension and diabetes mellitus [2]. The status of an AVF may vary based on the underlying disease in individuals with end-stage renal failure, and these differences could be reflected in the properties of AVF sounds. If the properties of AVF sounds associated with these diseases can be elucidated, the accuracy of AVF function assessment could be significantly improved.

Recent studies using machine learning to detect abnormalities in AVFs have demonstrated that AVF stenosis can be identified with high accuracy [16–19], but this is limited to the time of observation by clinicians at HD facilities. However, if HD patients can evaluate AVF function at home using AVF sounds, it may facilitate self-management to prevent AVF abnormalities.

Therefore, this study aimed to elucidate the relationships between the properties of AVF sounds and a medical/surgical history of HD recipients. This study is expected to aid patients in managing their AVFs and improving self-care practices by facilitating the assessment of AVF function. Furthermore, the findings of this study are anticipated to contribute to the development of advanced devices for detecting AVF failure.

## Materials and methods

In this study, a longitudinal survey of AVF sounds in individuals with AVFs was conducted. The period of investigation spanned from November 2017 to December 2020. This study focused on 53 of the 92 individuals who participated in a survey conducted at a dialysis facility in Prefecture A of Japan in 2017. The participants were undergoing HD via an AVF and provided consent to participate in the study.

### Data collection period

The AVF sounds were collected between November 15 and 30, 2017, and a medical/surgical history was collected between October 28 and November 15, 2020.

### Data collection methods and analyses

AVF sounds were obtained using a stethoscope manufactured by Focal Corporation (FC-200), a monaural microphone (AT9903) by Audio-Technica, and an IC recorder (DR-05) made by TASCAM [20]. The participants were seated in a quiet room, with their upper extremity positioned at chest level on an elbow pillow placed on a desk, as shown in Fig 1. During auscultation performed by the researcher, AVF sounds were captured using the stethoscope portion of the microphone, which was gently placed in close contact with the anastomosis of the AVF without applying pressure.

The recording duration was set to approximately 10–20 seconds, corresponding to 15–30 pulse cycles. To minimize noise caused by patient body movements, recordings were conducted only after confirming that the AVF sounds could be clearly auscultated and accurately captured.

In 2020, data on the participants' medical/surgical history were extracted from the medical records as of 2017, in which AVF sounds were initially recorded. The collected data included the time since AVF construction, medical history (e.g., diabetes mellitus and hypertension), events of surgical interventions on the AVF, and AVF reconstructions that occurred between 2017 and 2020 (henceforth, AVF-related events) (Fig 2).

### Analysis of AVF blood flow sounds

The recorded AVF sounds were initially processed using Audacity Ver3.0.0 (The Audacity Team, https://www.audacityteam.org/) to generate amplitude and power spectrograms. The power spectral density (PSD) was calculated via a fast Fourier transform implemented in MATLAB Version R2021a (MathWorks, Natick, MA, USA), a numerical analysis software package.

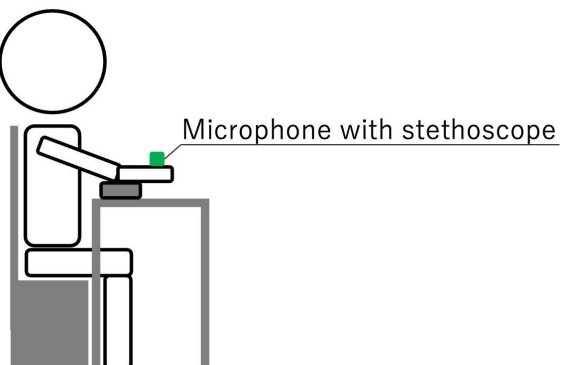

**Fig 1. Participant's body position when recording AVF sounds.** The patients place their upper limb at chest height on an elbow pillow on the desk, and the researcher records the arteriovenous fistula sounds.

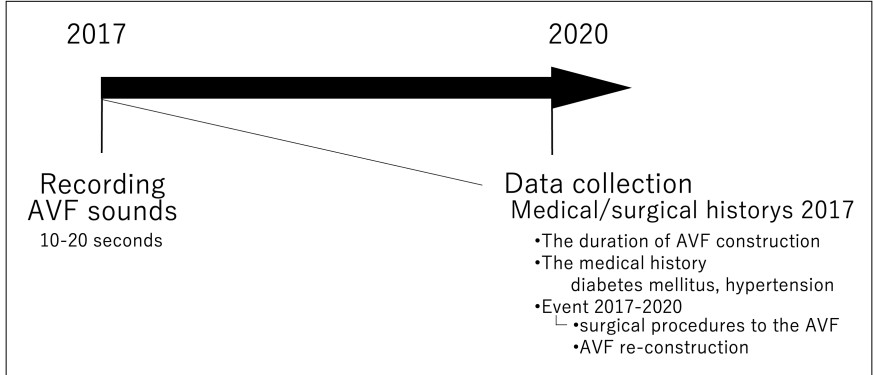

**Fig 2. Data collection method.** Arteriovenous fistula blood flow sounds were collected in 2017, and participants' medical/surgical history were collected in 2020. ᵃAVF sounds: arteriovenous fistula blood flow sounds.

### Extracting one cycle of AVF sounds

While listening to the AVF sounds, the segment corresponding to one cycle, from the start point to the end point, was extracted based on the amplitude and power spectrograms (Fig 3). This process was repeated three times for each recording. The duration of each extracted cycle was then measured.

### Identifying the point and frequency of the highest PSD in one cycle of AVF sounds

The following steps were performed to clarify the frequency properties of one cycle of AVF sounds:

1) PSD extraction: The PSD of one cycle of AVF sounds was extracted at 100-Hz intervals, spanning frequencies from 100 Hz to 4,000 Hz.

2) Normalization and Time at Maximum Power Output (TMP) calculation: The duration of each cycle of AVF sounds was normalized to 100 units. The time elapsed from the start of one cycle to the point where the PSD reached its maximum was expressed as a percentage of the one cycle's duration and referred to as TMP.

3) Maximum Power Output for each Frequency band (MF) Identification: The frequency band at which the PSD reached its maximum within one cycle of AVF sounds, in the range of 100–4000 Hz, was identified and referred to as MF.

Fig 4 shows the methods used for data collection and analysis.

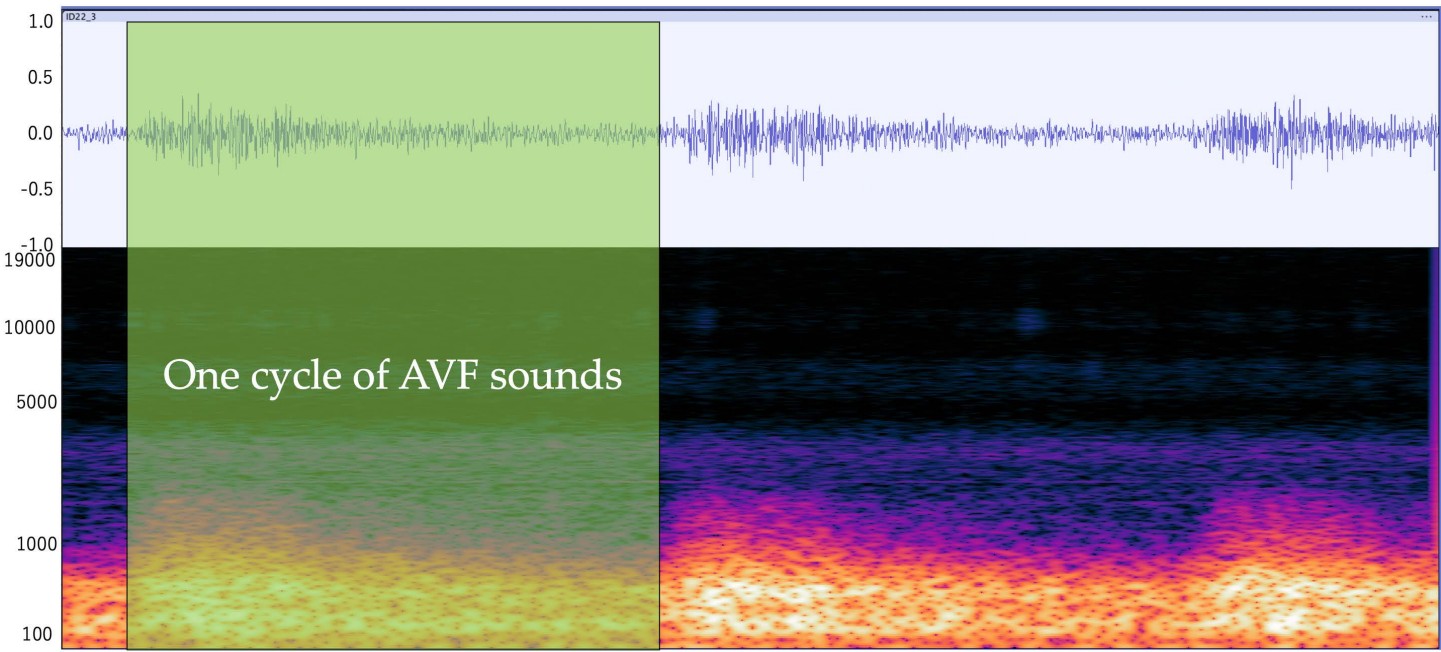

**Fig 3. Extracting one cycle of AVF sounds.**

## Statistical analysis

To clarify the factors affecting the properties of AVF sounds, regression analysis was performed using a generalized linear model (GLM) with the statistical analysis software R (version 4.3.1). The model incorporated logarithmic transformations for explanatory variables, which included the presence or absence of diabetes mellitus and hypertension, and the occurrence of AVF-related events. In addition, AVF vintage was categorized into two groups: less than 5 years, or 5 years or more. The response variable in the GLM was log (TMP).

$$log\ (TMP) = \beta 0 \cdot Hz + \beta 1 \cdot DM + \beta 2 \cdot HT + \beta 3 \cdot Event + \beta 4 \cdot Vintage + \beta 5 \qquad (1)$$

Definitions and summaries of variables are as follows.

Hz: Frequency range set to 100–700 Hz

TMP: The percentage of time within one cycle of AVF sounds required for the PSD to reach its maximum value

DM: Represents the presence of diabetes mellitus, coded as 1 for individuals with a pre-existing diagnosis and 0 for those without

HT: Represents the presence of hypertension, coded as 1 for individuals with a pre-existing diagnosis and 0 for those without

Event: Represents the occurrence of surgical procedures on AVFs, including AVF reconstruction, between 2017 and 2020. Coded as 1 if present and 0 if absent

Vintage: Represents the time since AVF construction. Based on previous studies indicating that the primary patency of AVFs is approximately 50% at 4 - 5 years [4,5], durations were categorized as follows: 1 for 5 years or more, and 0 for less than 5 years.

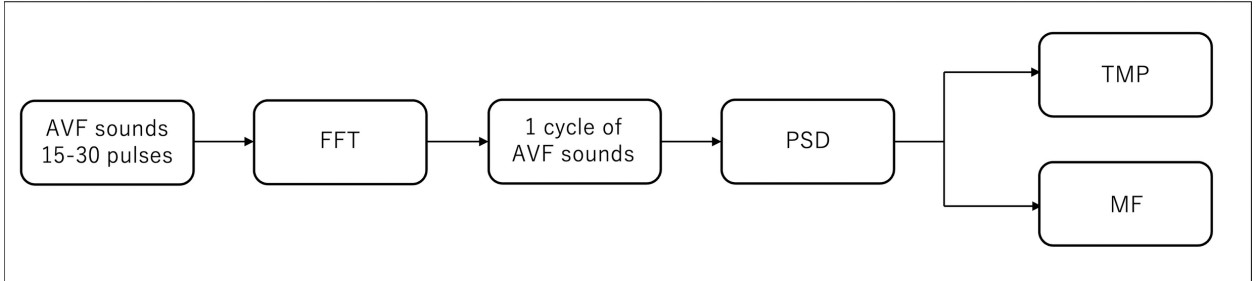

**Fig 4. Techniques for the analysis of AVF sounds.** [a] AVF sounds: arteriovenous fistula blood flow sounds. [b] PSD: power spectral density. [c] TMP: the percentage of time within one cycle of arteriovenous fistula sounds required for the power spectral density to reach its maximum value [d] MF: the frequency within one cycle of arteriovenous fistula sounds at which the power spectral density value reached its maximum in the 100-4000 Hz range.

Regression analysis for the model was performed using a Gaussian distribution with a log-link function. To examine the differences of AVF sounds according to the medical/surgical history, MF was used as the response variable. Explanatory variables included the presence of diabetes mellitus, hypertension, AVF-related events, and AVF vintage (<5 years or ≥5 years). Welch's *t-t*est was used to assess these differences, with the significance level set at 5%.

### Ethics statement

This study was conducted in compliance with the Declaration of Helsinki (2004) and was approved by the Ethics Committee of the authors' affiliated institution (approval numbers: 17–92 and 20–56). Prior to the initiation of the study, written, informed consent was obtained from all participants. Although the researcher had access to personal information during data collection (from October 28 to November 15, 2020), all data were anonymized at the time of collection to safeguard participant confidentiality and privacy.

## Results

Of the 53 participants, 29 (54.7%) had a documented history of diabetes mellitus, 36 (67.9%) had a history of hypertension, and 9 (17.0%) had an AVF-related event. The mean AVF vintage was 6.9±7.4 years. Of these participants, 22 individuals (41.5%) had an AVF vintage of less than 5 years, whereas 31 individuals (58.5%) had a vintage of 5 years or more (Table 1).

### Relationships between TMP and a medical/surgical history of the participants

The mean TMP values at each 100-Hz interval are shown in Fig 5. Across the frequency range of 100–4,000 Hz, the mean TMP was 29.2±16.7%, indicating that the values were consistently high, around 30%. The peak TMP, 30.4%, was recorded at 100 Hz. Between 200 Hz and 700 Hz, the TMP values gradually decreased to 24.6%. Beyond 800 Hz, the TMP values stabilized at approximately 30%.

Based on these observations, the analysis focused on the frequency range of 100–700 Hz. Using GLM, the investigation examined the medical/surgical history affecting TMP within this frequency range, leading to the development of Model A (Table 2).

Model A was

$$log\,(TMP) = -1.4774 - 0.0003Hz - 0.1714DM + 0.0068HT - 0.0340Event + 0.1309Vintage \qquad (2)$$

**Table 1. Overview of subjects (n = 53).**

| Medical/surgical history | | n | % |
|---|---|---|---|
| Diabetes mellitus | + | 29 | 54.7 |
| | − | 24 | 45.3 |
| Hypertension | + | 36 | 67.9 |
| | − | 17 | 32.1 |
| aEvent | + | 9 | 17.0 |
| | − | 44 | 83.0 |
| bVintage | <5 years | 22 | 41.5 |
| | ≥5 years | 31 | 58.5 |

aEvent: represents the occurrence of surgical procedures on arteriovenous fistulas, including arteriovenous fistula reconstruction, between 2017 and 2020.

bVintage: represents the time since arteriovenous fistula construction.

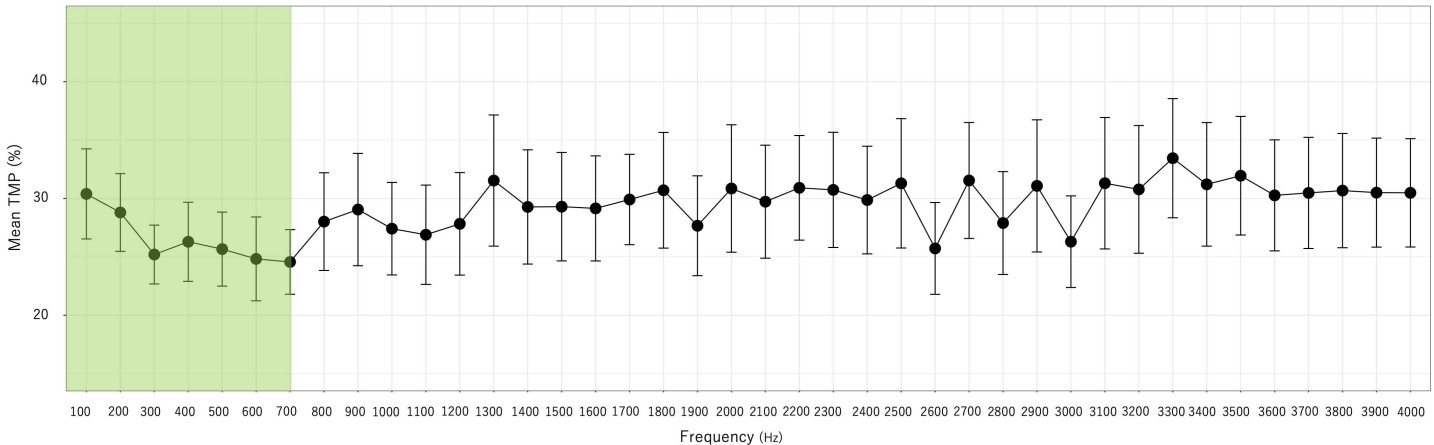

**Fig 5. Mean TMP in one cycle of AVF sounds (n = 53).** The focused region of 100–700 Hz is shown in green. Error bars indicate the 95% confidence interval.

The respective coefficients were as follows:-0.0003 (p = 0.002; 95% CI [-0.0006, -0.0001]) for Hz, -0.1714 (p < 0.001; 95% CI [-0.3429, -0.0797]) for DM, 0.0068 (p = 0.888; 95% CI [-0.0886, 0.1023]) for HT, -0.0340 (p = 0.574; 95% CI [-0.1525, 0.0845]) for event, and 0.1309 (p = 0.006; 95% CI [0.0378, 0.2240]) for AVF vintage. Significant associations were identified for Hz, DM, and AVF vintage, whereas HT and event showed no significant associations. Model B, which was derived using GLM and includes the significant variables Hz, DM, and vintage identified in Model A is shown in Table 3.

Model B was

$$log\,(TMP) = -1.4819 - 0.0003 Hz - 0.1709 DM + 0.1324 Vintage \tag{3}$$

The respective coefficients were as follows: -0.0003 (p = 0.002; 95% CI [-0.0006, -0.0001]) for Hz, -0.1709 (p < 0.001; 95% CI [-0.2624, -0.0795]) for DM, and 0.1324 (p = 0.005; 95% CI [0.0400, 0.2248]) for vintage. Significant associations were identified for Hz, DM, and vintage. As summarized in Tables 2 and 3, the Akaike information criterion (AIC) was

**Table 2. Medical/surgical history affecting TMP obtained from GLM (Model A).**

| Response variable | Predictor variable | aβ | bSE | c95% CI | | dP |
|---|---|---|---|---|---|---|
| TMP | Intercept | -1.4774 | 0.0651 | -1.6050 | -1.3498 | <0.0001 |
| | eHz | -0.0003 | 0.0001 | -0.0006 | -0.0001 | 0.002 |
| | Diabetes Mellitus | -0.1714 | 0.0468 | -0.3429 | -0.0797 | <0.0001 |
| | Hypertension | 0.0068 | 0.0487 | -0.0886 | 0.1023 | 0.888 |
| | fEvent | -0.0340 | 0.0605 | -0.1525 | 0.0845 | 0.574 |
| | gVintage | 0.1309 | 0.0475 | 0.0378 | 0.2240 | 0.006 |
| hAIC: 441.2 | | | | | | |

aβ: regression coefficient, bSE: standard error,

c95% CI: 95% confidence interval, dP: p–value, eHz: frequency,

fEvent: represents the occurrence of surgical procedures on arteriovenous fistulas, including arteriovenous fistula reconstruction, between 2017 and 2020

gVintage: represents the duration of arteriovenous fistula construction

hAIC: the Akaike information criterion

**Table 3. Medical/surgical history affecting TMP obtained from GLM (Model B).**

| Response variable | Predictor variable | aβ | bSE | c95% CI | | dP |
|---|---|---|---|---|---|---|
| TMP | Intercept | -1.4819 | 0.0618 | -1.6030 | -1.3608 | <0.0001 |
| | eHz | -0.0003 | 0.0001 | -0.0006 | -0.0001 | 0.002 |
| | Diabetes Mellitus | -0.1709 | 0.0467 | -0.2624 | -0.0795 | <0.0001 |
| | fVintage | 0.1324 | 0.0471 | 0.0400 | 0.2248 | 0.005 |
| gAIC: 444.8 | | | | | | |

aβ: regression coefficient,

bSE: standard error,

c95% CI: 95% confidence interval, dP: p–value, eHz: frequency,

fVintage: represents the time since arteriovenous fistula construction

gAIC: the Akaike information criterion

441.2 for Model A and 444.8 for Model B, indicating that Model B had a better fit. Based on Model B, TMP was found to decreased as frequency increased (Table 4).

TMP values were higher in AVFs with an AVF vintage of 5 years or more compared to those with an AVF vintage of less than 5 years, irrespective of diabetes mellitus status. At the same frequency, individuals without diabetes mellitus had higher TMP values than those with diabetes mellitus. For example, at 200 Hz, the TMP was estimated as follows: 18.0% (95% CI [15.1, 21.6%]), for individuals without diabetes mellitus and an AVF vintage of less than 5 years, 20.6% (95% CI [17.2, 24.7%]) for those without diabetes mellitus and an AVF vintage of 5 years or more, 21.4% (95% CI [17.9, 25.6%]) for those with diabetes mellitus and an AVF vintage of less than 5 years, and 24.4% (95% CI [20.4, 29.3%]) for those with diabetes mellitus and an AVF vintage of 5 years or more.

As shown in Fig 6, the PSD was significantly higher at 200 Hz than at other frequencies. Beyond 300 Hz, a decrease in PSD was observed, stabilizing with minimal variation at frequencies exceeding 800 Hz.

Given that the PSD at 200 Hz is distinctively higher, an index representing this characteristic PSD (PSD200) was calculated using the formula (200 Hz - 100 Hz) + (200 Hz - 300 Hz) based on the mean PSD values. Welch's t-test was subsequently performed to compare two patient groups categorized by an AVF vintage: less than 5 years and 5 years or more. The results are presented as mean values in Table 5.

**Table 4. Estimates of TMP by history of diabetes mellitus and AVF vintage at different frequencies in Model B.**

| | No history of diabetes mellitus | | | | History of diabetes mellitus | | | |
|---|---|---|---|---|---|---|---|---|
| AVF vintage | <5 years (n = 13) | | ≥5 years (n = 11) | | <5 years (n = 20) | | ≥5 years (n = 9) | |
| Frequency (Hz) | Estimate peak (%) | [a]95% CI | Estimate peak (%) | 95% CI | Estimate peak (%) | 95% CI | Estimate peak (%) | 95% CI |
| 100 | 18.6 | 15.5–22.3 | 21.2 | 17.7–25.4 | 22.0 | 18.4–26.4 | 25.2 | 21.0–30.2 |
| 200 | 18.0 | 15.1–21.6 | 20.6 | 17.2–24.7 | 21.4 | 17.9–25.6 | 24.4 | 20.4–29.3 |
| 300 | 17.5 | 14.6–21.0 | 20.0 | 16.7–23.9 | 20.8 | 17.3–24.9 | 23.7 | 19.8–28.4 |
| 400 | 17.0 | 14.2–20.3 | 19.4 | 16.2–23.2 | 20.2 | 16.8–24.1 | 23.0 | 19.2–27.6 |
| 500 | 16.5 | 13.8–19.7 | 18.8 | 15.7–22.5 | 19.6 | 16.3–23.4 | 22.3 | 18.6–26.7 |
| 600 | 16.0 | 13.4–19.2 | 18.3 | 15.2–21.9 | 19.0 | 15.8–22.7 | 21.7 | 18.1–26.0 |
| 700 | 15.5 | 13.0–18.6 | 17.7 | 14.8–21.2 | 18.4 | 15.4–22.1 | 21.0 | 17.6–25.2 |

[a]95%CI: 95% confidence interval

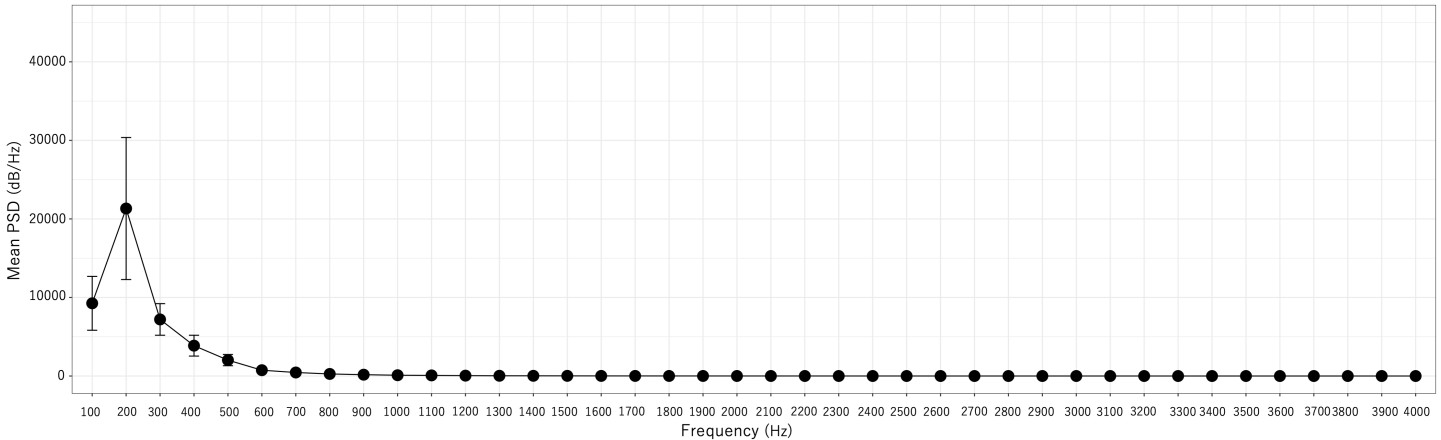

**Fig 6. Mean PSD for each frequency of AVF sounds (MF) (n = 53).** Error bars indicate the 95% confidence interval.

**Table 5. Comparison based on medical/surgical history of PSD 200.**

| Variable | | n | Mean (dB/Hz) | [a]SD | [b]95% CI | | [c]P |
|---|---|---|---|---|---|---|---|
| Diabetes mellitus | − | 24 | 39,052 | 74,987 | 7,388 | 70,717 | 0.158 |
| | + | 29 | 15,543 | 28,893 | 4,553 | 26,533 | |
| Hypertension | − | 17 | 15,840 | 47,758 | -8,715 | 40,394 | 0.321 |
| | + | 36 | 31,076 | 58,747 | 11,199 | 50,953 | |
| [d]Event | − | 44 | 30,011 | 60,147 | 11,724 | 48,298 | 0.022* |
| | + | 9 | 7,502 | 8,774 | 757 | 14,246 | |
| [e]Vintage | <5 years | 22 | 45,834 | 74,225 | 12,925 | 78,744 | 0.056 |
| | ≥5 years | 31 | 12,247 | 31,492 | 695 | 23,798 | |

[a]SD: standard deviation, [b]95% CI: 95% confidence interval, [c]P: p–value

[d]Event: represents the occurrence of surgical procedures on arteriovenous fistulas, including arteriovenous fistula reconstruction, between 2017 and 2020

[e]Vintage: represents the duration of arteriovenous fistula construction

*: Welch's *t*-test p-value < 0.05

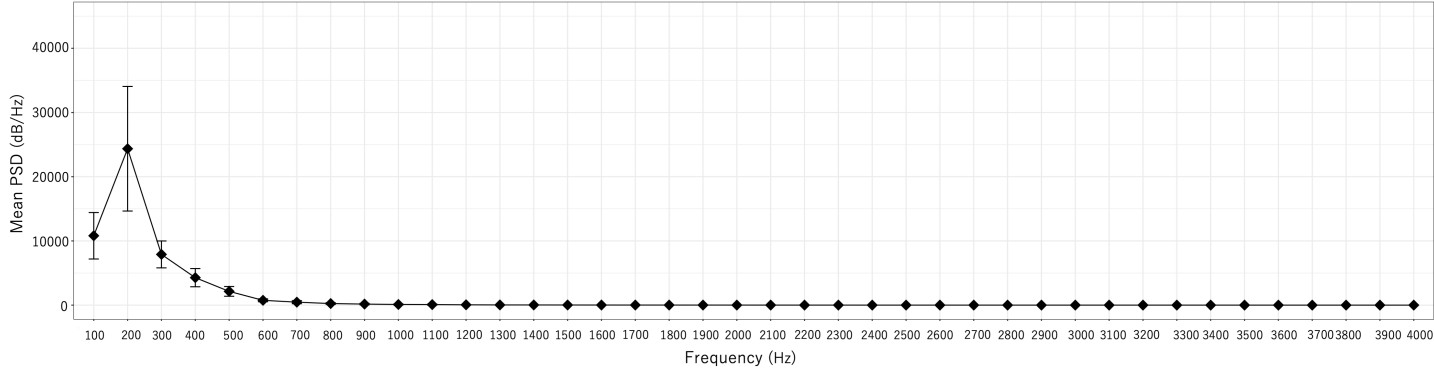

**Fig 7. Mean PSD for each frequency of AVF sounds (MF) in the no event group (n = 44).** Error bars indicate the 95% confidence interval.

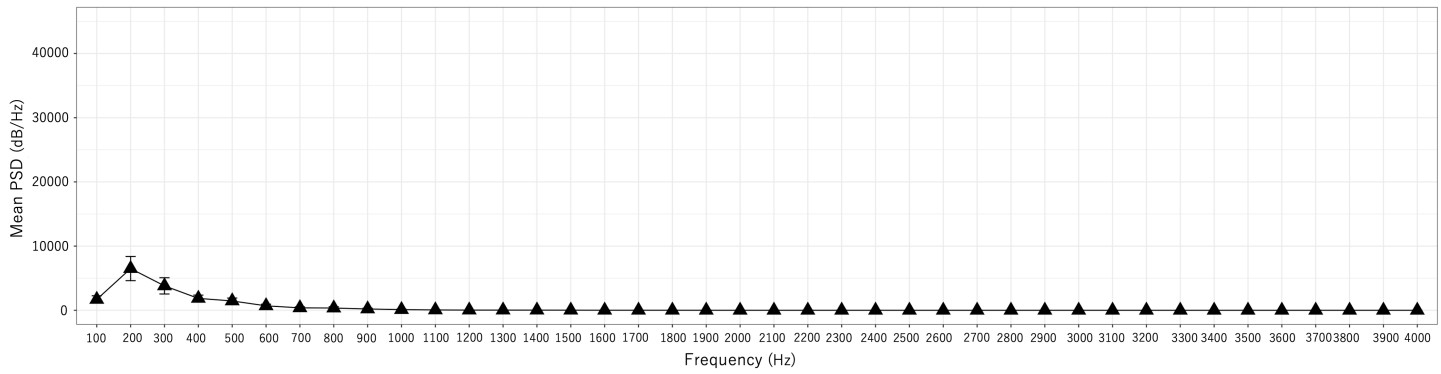

**Fig 8. Mean PSD for each frequency of AVF sounds (MF) in the event group (n = 9).** Error bars indicate the 95% confidence interval.

The mean PSD200 values were as follows: 39,052 (95% CI [7,388, 70,717]) in the no diabetes mellitus group and 15,543 (95% CI [4,553, 26,533]) in the diabetes mellitus group; 15,840 (95% CI [-8,715, 40,394]) in the no hypertension group and 31,076 (95% CI [11,199, 50,953]) in the hypertension group. For the event groups, the mean PSD200 was 30,011 (95% CI [11,724, 48,298]) in the no event group and 7,502 (95% CI [757, 14,246]) in the event group. Regarding AVF vintage, the PSD200 was 45,834 (95% CI [12,925, 78,744]) for AVFs with a vintage less than 5 years and 12,247 (95% CI [695, 23,798]) for AVFs with a vintage of 5 years or more.

A trend indicating smaller PSD values in the event group than in the other groups was observed. In addition, PSD200 was significantly higher in the no event group than in the event group (Figs 7 and 8).

## Discussion

This study analyzed the properties of AVF sounds by examining the time required for one cycle of AVF sounds as 100, the point of highest PSD (TMP), and the frequency band with the highest PSD at 100-Hz intervals (MF) within the range of 100–4,000 Hz.

Fluids, including blood, generate vortices when turbulence occurs in the flow. In fluid acoustics, it is well-established that fluid sounds are generated from the formation and dissipation of individual vortices, and the intensity of these sounds increases with higher flow velocity [21]. Therefore, the volume of AVF sounds is lower when the blood flow velocity in the AVF is low and high when velocity is high. Typically, during the isovolumetric contraction phase, there is minimal blood

ejection, resulting in reduced blood flow velocity. The velocity peaks during the rapid ventricular ejection phase and gradually decreases during the reduced ventricular ejection phase. Based on this, it is presumed that the TMP of a well-functioning AVF falls within the range corresponding to the rapid and reduced ventricular ejection phases of the cardiac cycle.

In the present study, patients without a history of diabetes mellitus and with an AVF vintage of less than five years demonstrated the lowest TMP values. Conversely, patients with a history of diabetes mellitus and an AVF vintage of five or more years exhibited the highest TMP values. According to previous studies, diabetes mellitus has been shown to contribute to arteriosclerosis [22], and AVFs in diabetic patients have lower patency rates than in those without diabetes mellitus [23,24]. Arteriosclerosis is considered a precursor to vascular stenosis [25], and the frequency of HD is associated with an increased risk of AVF failure [26,27]. The findings of the present, combined with prior research, suggest that diabetes mellitus-induced arteriosclerosis and a prolonged AVF vintage of five years or more negatively affect the vascular intima, resulting in higher TMP values. Therefore, combining TMP data with information on diabetes mellitus history and AVF vintage may provide valuable insights into the presence of AVF thickening or stenosis.

AVF stenosis is known to be associated with an increase in high-frequency components [13–15]. However, the PSD analysis conducted in the present study showed that consistently high values were seen at 200 Hz, regardless of the occurrence of AVF-related events within three years following the recording of AVF sounds. Further, PSD200, which is the difference between the PSD values at 200 Hz and those at 100 Hz and 300 Hz, was significantly lower in cases with AVF-related events than in those without. In the present study, the 95% confidence interval for PSD200 ranged from 757 to 14,246 in cases with an AVF-related event, and from 11,724–48,298 in cases without an AVF-related event; the term "event" refers to surgical interventions involving AVFs, including AVF reconstruction. AVF-related events are often attributed to complications such as AVF infection, pseudoaneurysm formation, venous hypertension, stenosis and occlusion [8]. AVF stenosis and occlusion are caused by turbulence resulting from the direct influx of blood from arteries to veins under high hemodynamic pressures. This turbulence induces thickening of the venous valve intima and narrowing of the vascular lumen. Additionally, repeated cannulations during HD contribute to the progressive weakening of the vascular access wall [28], further compromising its structural integrity. These factors collectively result in hypertrophy of the vascular intima [29], which is thought to reduce blood flow within the AVF and compromise its function. In cases with an AVF-related event, it is presumed that blood flow in the AVF was already reduced at the time of the AVF sound recording. This reduction in blood flow likely led to a decrease in the volume of the AVF sounds and a corresponding reduction in the PSD, which quantifies the intensity of the signal components of the AVF sounds. Supporting this, a previous study using the auditory spectrum flux and the auditory spectral centroid has demonstrated that the frequency components of a well-functioning AVF are predominantly concentrated within the range of 50–450Hz [30]. Furthermore, another earlier study using the Maximum Entropy Method has demonstrated that the frequency components of a well- functioning AVF are predominantly found below 500 Hz [31]. These previous findings support the result of the present study that the lower PSD200 observed in cases with an AVF-related event is due to a significant reduction in PSD within the frequency bands typically associated with a well-functioning AVF, and that a high PSD200 value may reflect a well-functioning AVF. In Japan, hemodialysis is typically performed three times per week [3], with AVFs allowing direct blood flow from arteries to veins under similar conditions. In addition to hypertension and diabetes mellitus, which were examined in the present study, individuals experiencing AVF-related events may have other risk factors contributing to intimal hyperplasia, such as hyperlipidemia, obesity, and smoking. Although further investigation is required to identify these factors comprehensively, this study shows that a PSD200 value below 14,246 appears to indicate an elevated risk of AVF failure.

In the present study, the status of the AVF was not evaluated using ultrasonography or angiography. Therefore, assessing the objective risk of AVF failure beyond the acoustic analysis of blood flow remains a critical area for future research. The present study involved 53 participants, of whom only 9 underwent AVF-related interventions between 2017 and 2020. Consequently, further validation with a larger sample size is necessary to generalize the findings of this study.

## Conclusions

To elucidate the relationships between the properties of AVF sounds and a medical/surgical history of HD patients, this study analyzed the PSD of AVF sounds across different frequencies. The findings showed that the PSD of a well-functioning AVF was the highest within the range of 0–25% of one cycle of AVF sounds, with 200 Hz identified as the frequency the highest PSD within the range of 100–4,000Hz.

These results indicate the potential of using AVF sound analysis to identify HD patients at high risk of AVF failure.

## Supporting information

**S1 Data. Overview of subjects.**
(XLSX)

**S2 Data. Mean TMP in vessel flow sounds of one cycle of AVF sounds(n = 53).**
(XLSX)

**S3 Data. Mean PSD for each frequency of AVF blood flow sounds (MF)(n = 53).**
(XLSX)

**S4 Data. Comparison based on physical characteristics of PSD 200.**
(XLSX)

**S5 Data. Mean PSD for each frequency of arteriovenous fistula blood flow sounds (MF) in the no Event group (n = 44).**
(XLSX)

**S6 Data. Mean PSD for each frequency of arteriovenous fistula blood flow sounds (MF) in the Event group (n = 9).**
(XLSX)

## Acknowledgments

The authors would like to thank the patients and staff of Tsukagawa Daiichi Hospital and Daisuke Higashi for their cooperation in this study.

## Author contributions

**Conceptualization:** Keiko Tanaka.

**Data curation:** Keiko Tanaka.

**Formal analysis:** Keiko Tanaka, Satoko Shin, Michiaki Kai.

**Funding acquisition:** Keiko Tanaka.

**Investigation:** Keiko Tanaka, Keisuke Nishijima.

**Methodology:** Keiko Tanaka, Keisuke Nishijima, Ken'ichi Furuya, Satoko Shin, Michiaki Kai.

**Project administration:** Keiko Tanaka.

**Resources:** Keiko Tanaka.

**Software:** Keiko Tanaka.

**Supervision:** Keiko Tanaka.

**Validation:** Keiko Tanaka, Satoko Shin, Michiaki Kai.

**Visualization:** Keiko Tanaka, Satoko Shin, Michiaki Kai.

**Writing – original draft:** Keiko Tanaka.

**Writing – review & editing:** Keiko Tanaka, Satoko Shin, Michiaki Kai.

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
