## [Decision Letter · Decision Letter 0]

22 Nov 2024

PONE-D-24-46595Risk assessment of arteriovenous fistulas using arteriovenous fistula blood flow sounds and physical characteristics of hemodialysis patientsPLOS ONE

Dear Dr. Tanaka,

Thank you for submitting your manuscript to PLOS ONE. After careful consideration, we feel that it has merit but does not fully meet PLOS ONE’s publication criteria as it currently stands. Therefore, we invite you to submit a revised version of the manuscript that addresses the points raised during the review process.

The manuscript is potentially interesting, but some revisions are needed before we can reconsider it again.

We look forward to receiving your revised manuscript.

Kind regards,

Raffaele Serra, M.D., Ph.D

Academic Editor

PLOS ONE

“This work was supported by JSPS KAKENHI Grant Numbers JP17K17415, JP21K10775.”

Reviewers' comments:

Reviewer's Responses to Questions

**Comments to the Author**

1. Is the manuscript technically sound, and do the data support the conclusions?

Reviewer #1: Yes

2. Has the statistical analysis been performed appropriately and rigorously? 

Reviewer #1: Yes

3. Have the authors made all data underlying the findings in their manuscript fully available?

Reviewer #1: Yes

4. Is the manuscript presented in an intelligible fashion and written in standard English?

Reviewer #1: Yes

5. Review Comments to the Author

Reviewer #1: The aim of this paper is to clarify the relationships between the characteristics of AVF sounds and the physical characteristics of HD patients, a longitudinal survey of AVF sounds was conducted in patients with autologous AVF. It is very interesting but are necessary some revisions:

1) I suggest to include the following paper: https://doi.org/10.3390/life14111382;

2) The language must be reviewed in details;

3) The abstract is not clear, I suggest to improve the structure;

4) Methods are clear but the language must be improved;

5) Discussion must be reviewed with a comparation between your results and similar research.

6. PLOS authors have the option to publish the peer review history of their article (what does this mean? ). If published, this will include your full peer review and any attached files.

**Do you want your identity to be public for this peer review?** For information about this choice, including consent withdrawal, please see our Privacy Policy .

Reviewer #1: No

---

## [Author Response · Author response to Decision Letter 0]

11 Jan 2025

Dear Dr. Raffaele Serra,

Thank you very much for reviewing our manuscript. Below, we have addressed the points you raised. The same information has also been included in the "Response to Reviewers."

Thank you for your guidance regarding PLOS ONE's style requirements. We have carefully reviewed and revised our manuscript to ensure it aligns with the journal's guidelines, including those pertaining to file naming conventions. All revisions have been highlighted in blue for your convenience. If there are any additional areas that require further adjustments, we respectfully request your feedback.

•We started a new line numbering sequence from the Abstract onward.

•Annotations for figures and tables have been marked with symbols, and new captions have been added.

We have included the following statement in the cover letter to address this requirement:

3. Please include captions for your Supporting Information files at the end of your manuscript, and update any in-text citations to match accordingly.

After careful review, we confirm that our manuscript does not include any Supporting Information files, as all necessary data, figures, and tables are presented within the main text. No additional captions or citations are required. We have reviewed the manuscript to ensure clarity and to avoid any potential confusion regarding supplementary materials.

Sincerely,

Authors

Dear Reviewer

Thank you for taking the time to review our manuscript. Below, we have addressed the points you raised. The same information has also been included in the "Response to Reviewers."

1. I suggest to include the following paper:https://doi.org/10.3390/life14111382.

We have incorporated the content of the suggested paper in lines 39- 41 of the manuscript. Further, to align with this inclusion, we have added a related statement to lines 44-45. The added content is highlighted in red in the "Revised Manuscript with Track Changes" file.

2. The language must be reviewed in details.

The language throughout the manuscript has been thoroughly reviewed and revised to improve clarity and readability. Since the revisions cover the entire manuscript, all changes have been highlighted in green in the "Revised Manuscript with Track Changes" file. We hope the revised manuscript addresses your concerns effectively.

3. The abstract is not clear, I suggest to improve the structure.

We have completely revised the abstract to improve its clarity, structure, and alignment with the study's objectives and findings. The revised version provides a clear and concise summary of the study's background, methods, results, and conclusions. We hope that the updated abstract addresses your concerns effectively.

4. Methods are clear but the language must be improved.

As noted in Comment 2, the "Methods" section has undergone a thorough language review and revisions to enhance its readability. Since these revisions extend throughout the entire "Methods" section, all changes have been highlighted in green in the "Revised Manuscript with Track Changes" file for your reference.

5. Discussion must be reviewed with a comparation between your results and similar research.

To address this comment, we have incorporated comparisons with relevant studies in the Discussion section. While we searched for prior studies directly associating AVF sounds with physical characteristics, we could not identify any. Therefore, we referred to literature demonstrating associations between diabetes and AVF patency rates and made the following adjustments to Lines 330-338.

Sincerely,

Authors

---

## [Decision Letter · Decision Letter 1]

9 Mar 2025

PONE-D-24-46595R1Risk assessment of arteriovenous fistulas using arteriovenous fistula blood flow sounds and physical characteristics of hemodialysis patientsPLOS ONE

Dear Dr. Tanaka,

Thank you for submitting your manuscript to PLOS ONE. After careful consideration, we feel that it has merit but does not fully meet PLOS ONE’s publication criteria as it currently stands. Therefore, we invite you to submit a revised version of the manuscript that addresses the points raised during the review process.

We look forward to receiving your revised manuscript.

Kind regards,

Utpal Sen, Ph.D.

Academic Editor

PLOS ONE

Journal Requirements:

Reviewers' comments:

Reviewer's Responses to Questions

**Comments to the Author**

1. If the authors have adequately addressed your comments raised in a previous round of review and you feel that this manuscript is now acceptable for publication, you may indicate that here to bypass the “Comments to the Author” section, enter your conflict of interest statement in the “Confidential to Editor” section, and submit your "Accept" recommendation.

Reviewer #1: (No Response)

2. Is the manuscript technically sound, and do the data support the conclusions?

Reviewer #1: (No Response)

3. Has the statistical analysis been performed appropriately and rigorously? 

Reviewer #1: (No Response)

4. Have the authors made all data underlying the findings in their manuscript fully available?

Reviewer #1: (No Response)

5. Is the manuscript presented in an intelligible fashion and written in standard English?

Reviewer #1: (No Response)

6. Review Comments to the Author

Reviewer #1: To maintain the arteriovenous fistula (AVF) of hemodialysis (HD) patients in a condition

that allows for long-term HD, there is a need to develop a method to easily assess AVF

function from any location. Therefore, to clarify the relationships between the

characteristics of AVF sounds and the physical characteristics of HD patients, a

longitudinal survey of AVF sounds was conducted in patients with autologous AVF.

Using the fast Fourier transform, the power spectral density (PSD) of AVF blood flow

sounds was calculated.

The paper is well structured but I suggest to include the following paper: https://doi.org/10.3390/life14111382 and I suggest to revise the language.

7. PLOS authors have the option to publish the peer review history of their article (what does this mean? ). If published, this will include your full peer review and any attached files.

**Do you want your identity to be public for this peer review?** For information about this choice, including consent withdrawal, please see our Privacy Policy .

Reviewer #1: No

---

## [Author Response · Author response to Decision Letter 1]

11 Mar 2025

To the Editor, Dr. Utpal Sen,

We sincerely appreciate your valuable insights regarding the references. We have carefully revised the manuscript and the reference list accordingly. We hope that our submission now fully meets the publication standards of PLOS ONE. Thank you for your time and consideration.

To the Reviewer,

Thank you for suggesting the relevant paper. We have revised the wording in the Introduction and incorporated the suggested reference accordingly. We appreciate your time and effort in reviewing our manuscript and look forward to your feedback.

Sincerely,

Keiko Tanaka

---

## [Decision Letter · Decision Letter 2]

10 Apr 2025

PONE-D-24-46595R2Risk assessment of arteriovenous fistulas using arteriovenous fistula blood flow sounds and physical characteristics of hemodialysis patientsPLOS ONE

Dear Dr. Tanaka,

Thank you for submitting your manuscript to PLOS ONE. After careful consideration, we feel that it has merit but does not fully meet PLOS ONE’s publication criteria as it currently stands. Therefore, we invite you to submit a revised version of the manuscript that addresses the points raised during the review process.

We look forward to receiving your revised manuscript.

Kind regards,

Utpal Sen, Ph.D.

Academic Editor

PLOS ONE

Journal Requirements:

**Additional Editor Comments:**

Take professional help to revise / edit the language.

Reviewers' comments:

Reviewer's Responses to Questions

**Comments to the Author**

1. If the authors have adequately addressed your comments raised in a previous round of review and you feel that this manuscript is now acceptable for publication, you may indicate that here to bypass the “Comments to the Author” section, enter your conflict of interest statement in the “Confidential to Editor” section, and submit your "Accept" recommendation.

Reviewer #1: All comments have been addressed

2. Is the manuscript technically sound, and do the data support the conclusions?

Reviewer #1: Yes

3. Has the statistical analysis been performed appropriately and rigorously? 

Reviewer #1: Yes

4. Have the authors made all data underlying the findings in their manuscript fully available?

Reviewer #1: Yes

5. Is the manuscript presented in an intelligible fashion and written in standard English?

Reviewer #1: No

6. Review Comments to the Author

Reviewer #1: This study aimed to clarify the

relationship between the sound properties of blood flow through arteriovenous fistulas

and the physical characteristics of patients undergoing hemodialysis by analyzing the

distribution of sound frequencies from 100 to 4,000 Hz. The paper is interesting but I suggest to revise the language.

7. PLOS authors have the option to publish the peer review history of their article (what does this mean? ). If published, this will include your full peer review and any attached files.

**Do you want your identity to be public for this peer review?** For information about this choice, including consent withdrawal, please see our Privacy Policy .

Reviewer #1: No

---

## [Author Response · Author response to Decision Letter 2]

23 Apr 2025

Thank you very much for your valuable comments regarding the language. We have had the manuscript professionally revised by experts in both English and our native language. We kindly ask for your review and confirmation.

---

## [Editor Report · Decision Letter 3]

30 Apr 2025

Risk assessment of arteriovenous fistulas focusing on the relationships between the properties of shunted blood flow sounds and a medical/surgical history of hemodialysis patients

PONE-D-24-46595R3

Dear Dr. Tanaka,

We’re pleased to inform you that your manuscript has been judged scientifically suitable for publication and will be formally accepted for publication once it meets all outstanding technical requirements.

Kind regards,

Utpal Sen, Ph.D.

Academic Editor

PLOS ONE
---

## [Editor Report · Acceptance letter]

PONE-D-24-46595R3

PLOS ONE

Dear Dr. Tanaka,

I'm pleased to inform you that your manuscript has been deemed suitable for publication in PLOS ONE. Congratulations! Your manuscript is now being handed over to our production team.

Kind regards,

on behalf of

Dr. Utpal Sen

Academic Editor

PLOS ONE